# Manipulating exchange bias in 2D magnetic heterojunction for high-performance robust memory applications

Xinyu Huang[1,2,7], Luman Zhang[3,7], Lei Tong[1,7], Zheng Li[1], Zhuiri Peng[1], Runfeng Lin[1], Wenhao Shi[1], Kan-Hao Xue [1], Hongwei Dai[3], Hui Cheng [3], Danilo de Camargo Branco [4], Jianbin Xu [5], Junbo Han [3] ✉, Gary J. Cheng [4] ✉, Xiangshui Miao [1,2] ✉ & Lei Ye [1,2,6] ✉

The exchange bias (EB) effect plays an undisputed role in the development of highly sensitive, robust, and high-density spintronic devices in magnetic data storage. However, the weak EB field, low blocking temperature, as well as the lack of modulation methods, seriously limit the application of EB in van der Waals (vdW) spintronic devices. Here, we utilized pressure engineering to tune the vdW spacing of the two-dimensional (2D) $FePSe_3/Fe_3GeTe_2$ heterostructures. The EB field ($H_{EB}$, from 29.2 mT to 111.2 mT) and blocking temperature ($T_b$, from 20 K to 110 K) are significantly enhanced, and a highly sensitive and robust spin valve is demonstrated. Interestingly, this enhancement of the EB effect was extended to exposed $Fe_3GeTe_2$, due to the single-domain nature of $Fe_3GeTe_2$. Our findings provide opportunities for the producing, exploring, and tuning of magnetic vdW heterostructures with strong interlayer coupling, thereby enabling customized 2D spintronic devices in the future.

Natural two-dimensional (2D) magnetic crystals and related van der Waals (vdW) heterostructures are premium candidates for studying novel magnetic phenomena and realizing innovative device structures[1–8]. In particular, ferromagnets-based vdW heterostructures constructed from various 2D materials have exhibited interesting properties and functionalities[9–13], which offer great potential in nanoscale spintronics applications[14–17]. A central research goal in 2D spintronics lies in developing effective methods for generating, transmitting, and detecting spin information based on 2D vdW materials. The exchange bias (EB) effect, for which the spins of a ferromagnet (FM) are pinned by those of an antiferromagnet (AFM)[18–20], has become an integral part of modern magnetism and is essential to this goal, as it provides a well-defined principal direction of spin

polarization for spintronic devices. Especially, fundamental research interests and numerous device applications have widely embedded 2D spin valves and 2D MTJs for in-memory technologies such as storage media, readout sensors, and magnetic random-access memory (MRAM)[7,21,22].

Recently, the EB effect of magnetic 2D vdW heterostructures has been studied in $CrCl_3/Fe_3GeTe_2$[23], $MnPS_3/Fe_3GeTe_2$[24], $MPSe_3/Fe_3GeTe_2$[25], $FePS_3/Fe_5GeTe_2$[26], and $Fe_3GeTe_2$[27] systems. However, EB effect of 2D vdW heterostructures faces challenges such as weak EB field $H_{EB}$ and low blocking temperature $T_b$, because the existence of vdW interface gap and interfacial contamination in the 2D heterostructure tend to yield weak interlayer coupling, which cannot provide a sufficient EB field[28–30]. Consequently, it is crucial to effectively

[1]School of Integrated Circuits, Wuhan National Laboratory for Optoelectronics, Huazhong University of Science and Technology, Wuhan 430074, China. [2]Hubei Yangtze Memory Laboratories, Wuhan 430205, China. [3]Wuhan National High Magnetic Field Center and Department of Physics, Huazhong University of Science and Technology, Wuhan 430074, China. [4]School of Industrial Engineering and Birck Nanotechnology Centre, Purdue University, West Lafayette, IN 47907, USA. [5]Department of Electronic Engineering, Materials Science and Technology Research Center, The Chinese University of Hong Kong, Hong Kong, China. [6]State Key Laboratory of Infrared Physics, Shanghai Institute of Technical Physics Chinese Academy of Sciences, Shanghai 200083, China. [7]These authors contributed equally: Xinyu Huang, Luman Zhang, Lei Tong. ✉e-mail: junbo.han@mail.hust.edu.cn; gjcheng@purdue.edu; miaoxs@hust.edu.cn; leiye@hust.edu.cn

enhance interlayer coupling for the purpose of a stronger EB effect in 2D heterostructure-based spintronic devices. Specifically, the sensitivity to interlayer coupling enables effective tuning of material properties through external modulation of the vdW interface distance[31–33]. Thus, manipulating the interlayer vdW spacing in magnetic 2D vdW heterostructures has been considered an effective way towards exchange coupling enhancement[16,34,35], but was paid with less attention thus far. To obtain strong magnetic coupling rapidly without damage by manipulating the vdW interface spacing of magnetic vdW heterostructures over large areas remains a challenge, which is a key to the observation and application of the EB effect in many new spintronic devices.

In this work, we demonstrated a proximity-induced EB effect in vdW magnetic heterostructures formed by $Fe_3GeTe_2$ (FGT) and $FePSe_3$ (FPSe), which can be effectively modulated via vdW interface spacing turning with the aid of laser shocking engineering (LS). After controllably applying high-pressure shocking in an ultra-short time (few picoseconds), the vdW spacing can be permanently tuned to enhance its interlayer coupling. The enhanced interlayer coupling of the FPSe/FGT heterostructure led to an impressive improvement of the EB field ($H_{EB}$, from 29.2 to 111.2 mT) and the blocking temperature ($T_b$, 110 K near the Néel temperature of FPSe (113 K)). In addition, a high-quality tunneling spin valve (FPSe/FGT/h-BN/FGT) was fabricated and investigated. After LS, the enhanced magnitude of the tunneling magnetoresistance (TMR) of 154% and the field window of 320 mT at 5 K was observed, respectively. The field window is around 15 times larger than the vertical FGT/h-BN/FGT spin valves before LS in our work. Interestingly, the exposed region of FGT connected to the heterostructure (connected FGT covered by FPSe) showed a similar improvement of $H_{EB}$ and $T_b$, completely contrary to the results measured from bare FGT that was isolated from the heterostructure. This phenomenon is attributed to the single-domain nature of ferromagnetic FGT. Our findings offer new insights for regulating the AFM/FM interlayer coupling through pressure engineering and show the possibility of developing novel and extensive vdW heterostructures with adjustable interlayer coupling. Furthermore, the robust and sizable EB effect for vdW magnets persisting up to relatively high temperatures presents a significant advance for realizing practical next-generation 2D spintronics devices.

## Results

### VdW heterostructure interlayer spacing regulation via laser shocking

A typical MRAM based on the EB effect is composed of a matrix of spin valves of vdW heterostructures. To obtain higher performance requirements such as high tunneling magnetoresistance and high read fault tolerance[5,7], the modulation of the EB effect can enhance the interlayer coupling to optimize performance. As shown in Fig. 1a, according to the generalized Meiklejohn-Bean model[36], $H_{EB}$ is inversely proportional to the interfacial spacing (Fig. 1aI). Therefore, reducing the interfacial spacing of the AFM/FM heterostructure can induce a stronger EB field and saturation magnetization (Fig. 1a II). Here, FGT and FPSe single crystals were confirmed by scanning electron microscope and energy dispersive spectrometer (Supplementary Figs. 1 and 2), and their bulk magnetizing characteristics are shown in Supplementary Fig. 3, indicating a Néel temperature $T_N$ of 113 K for FPSe and a Curie temperature $T_C$ of 230 K for FGT, consistent with those reported in the literature[37–40]. To study the interlayer coupling tuned by vdW spacing engineering, we prepared heterostructures composed of FGT and FPSe flakes. Raman spectrum characterized the FPSe, FGT, and FPSe/FGT heterostructure, where the peaks could well match those of FPSe and FGT, indicating the excellent quality of the FPSe/FGT heterostructure (Supplementary Fig. 4). The LS technology enables the modulation of the interlayer spacing of the vdW heterostructures, which can provide high pressure up to ~20 GPa peak level in an ultra-

short time scale (tens of picosecond), like a "hammering" operation. Figure 1b II shows a schematic diagram of the LS process (see the "Methods" for details). As shown in Fig. 1b, the FPSe/FGT vdW heterostructure with a natural vdW interface distance of $d_1$ exhibits a weak interlayer coupling. After LS[41,42], its vdW interface distance was reduced to $d_2$, promising a stronger interlayer coupling. The evolution of interface spacing under LS was studied by molecular dynamics (MD) simulations (Fig. 1c). The equilibrium states for both the unstrained configuration and strained configuration were simulated for the FPSe/FGT system, showing an interface spacing (distance between flake edge atoms) of 6.6 Å and 4.2 Å before and after LS, respectively (Supplementary Information Sections 3 and 4). Supplementary Figs. 5–7 and Tables 1–3 show the calculation details and strain evolution of the LS MD simulation. More importantly, LS is effective to flatten wrinkles and voids without any damage to the heterostructures[43,44]. The Raman spectrum and mapping before and after LS are shown in Supplementary Figs. 8 and 9, the $E_{2g}^2$ peak of FGT, and the $E_{2g}^1$ peak of FGT and $A_{1g}$ peak of FPSe in the FPSe/FGT heterostructure showed negligible change, indicating that FGT and FPSe/FGT heterostructure sustained their good quality after LS. Further, we used cross-sectional (scanning transmission electron microscopy) STEM images to measure the interface gap width of the FPSe/FGT heterostructure. The STEM images of the samples before and after LS were acquired along different orientations because the laser shocking process outside the chamber made it hard to conduct in situ characterizations. After LS, vdW interlayer distances of FPSe and FGT were fixed at their intrinsic values of ~2.5 and ~2.1 Å, respectively, consistent with previous reports[40,45–47], and there was no damage to the lattice structure, but the interface gap width (initial ~7.4 Å) was dramatically reduced to ~4.6 Å (Fig. 1d, e).

### Enhanced interfacial magnetic coupling by interlayer spacing modulation

According to the above mechanism, reducing the interface spacing in AFM/FM heterostructures is expected to significantly enhance $H_{EB}$ and $T_b$. To confirm the enhancement of $H_{EB}$ and $T_b$ in AFM/FM heterostructures via LS technology, magneto-optical Kerr effect (MOKE) techniques were used to investigate the magnetic properties of a FPSe/FGT heterostructure, denoted as sample A (Supplementary Fig. 10a). Sample A was characterized by atomic force microscope, showing the FPSe thicknesses of ~24.4 nm and the FGT thicknesses of ~18.0 nm (Supplementary Fig. 10b). We first explored the relationship of Kerr rotation angle ($\theta_k$) with respect to the magnetic field of FPSe/FGT before LS. Figure 2a shows the typical temperature-dependent Kerr loops of FPSe/FGT as a function of magnetic field (B) from 5 K to 80 K, where $\theta_k$ vs B exhibits an obvious shift along the negative B direction, signifying the emergence of $H_{EB}$. However, the typical MOKE curves of FGT before LS as a function of magnetic field (B) from 5 K to 200 K showed that their Kerr hysteresis loops remained symmetric relative to the zero point along the B-axis, suggesting a lack of $H_{EB}$ (Supplementary Fig. 11). The EB effect could be explained by the Meiklejohn-Bean model (namely M-B model, FPSe and FGT both with a single-domain state). In the AFM/FM system, an adjacent AFM layer causes a unidirectional pinning of the FM, manifesting as a shift in the hysteresis loop along the magnetic field axis below the Néel temperature ($T_N$) of the AFM, known as the EB field $H_{EB}$ (Supplementary Fig. 12). With the increase of temperature, $H_{EB}$ decreases gradually and disappears above 20 K, indicating a $T_b$ of EB field at 20 K.

On the other hand, according to the generalized Meiklejohn-Bean model[48–50], $H_{EB}$ can be written as:

$$H_{EB} = -\frac{A_{12}/\xi}{\mu_0 M_{FM} t_{FM}} + \frac{A_{12}^3}{8K_{AFM}^2 \mu_0 M_{FM} t_F t_{AFM}^2} \quad (1)$$

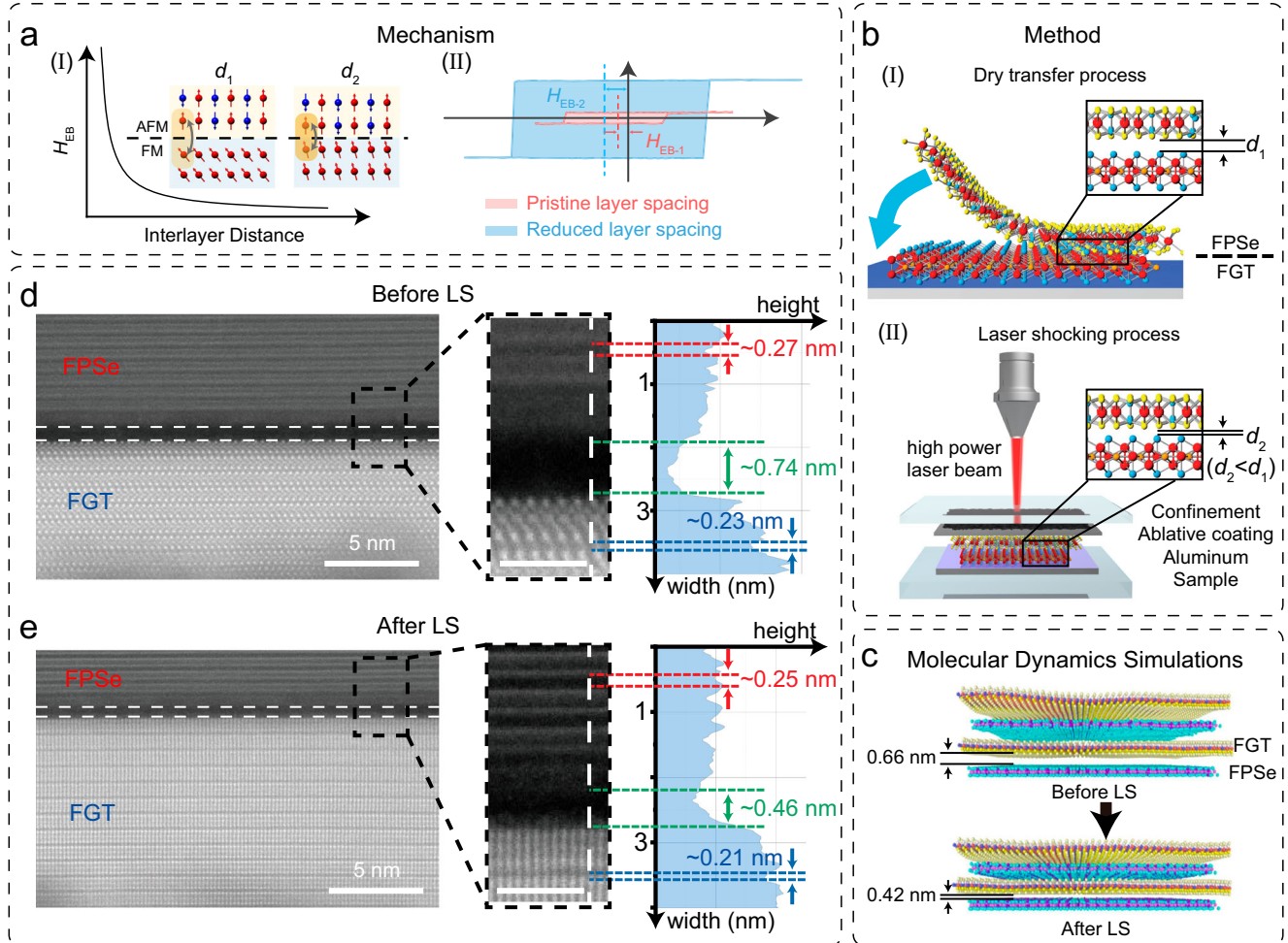

**Fig. 1 | Characterization of vdW spacing of FPSe/FGT heterostructures. a** Mechanism and schematic diagram of EB effect enhanced by reducing the AFM/FM interlayer spacing. **b** The schematic of the transfer process of FPSe/FGT heterostructure (I) and LS process (II) on a SiO2/Si substrate ($d_1 > d_2$). Panel II shows a schematic diagram of the laser shocking process. **c** Equilibrium position for a FPSe/ FGT system at the unstrained condition (top), strained condition (bottom) in molecular dynamics simulations. **d, e** The cross-sectional HAADF-STEM image of the FPSe/FGT heterostructure before (**d**) and after (**e**) LS. Inset: intensity profile (right panel) along white dashed line in the cross-sectional HAADF-STEM image (middle panel).

Where $A_{12}$ is the interfacial exchange stiffness, $\xi$ is the distance of the interlayer, $M_{FM}$ is the saturation magnetization of the ferromagnet, $K_{AFM}$ is the anisotropy constant of the AFM layer, $t_{FM}$ is the FM thickness, and $t_{AFM}$ is the AFM thickness. Equation (1) reveals that vdW spacing engineering can manipulate the magnetic coupling in vdW FPSe/FGT heterostructure. Therefore, we utilized LS to perform vdW spacing engineering, inducing extreme out-of-plane pressure onto another vdW FPSe/FGT heterostructure. Supplementary Table 4 provides a relationship between laser power and pressure in Supplementary Information Section 5. Its EB effect was effectively enhanced, as shown in Fig. 2b. The $H_{EB}$ and $T_b$ of sample A before and after LS are shown in Fig. 2c, d. After LS, $H_{EB}$ increased from 29.1 to 111.2 mT at 5 K, and $T_b$ increased from 20 to 110 K, which is close to the Néel temperature of FPSe (113 K), as shown in Fig. 2e. The prominent improvement of $H_{EB}$ and $T_b$ confirms that the interlayer coupling of 2D magnetic heterostructures can be strongly enhanced by LS. Furthermore, Fig. 2f, g demonstrates the significant enhancement of the coercive field ($H_{C-L}$: left coercive field, $H_{C-R}$: right coercive field), which can also be verified from Fig. 2h. In Fig. 2h, to further confirm the enhancement of the coercive field, an on-off ratio parameter $R_C$ is defined as $R_C = \frac{\theta_{ku(-180mT)} - \theta_{kd(-180mT)}}{\theta_{ku(0mT)} - \theta_{kd(0mT)}}$, where $\theta_{ku(-180\,mT)}$, $\theta_{kd(-180\,mT)}$, $\theta_{ku(0\,mT)}$, and $\theta_{kd(0\,mT)}$ represent the magneto-optical Kerr angles at −180 and 0 mT, corresponding to up and down sweeping directions of the hysteresis loop, respectively. The significantly enhanced coercive field offers a remarkable advantage in obtaining a large field window to reduce the error rate of data writing and reading during spin valves and MTJs design.

## Comparison between interlayer spacing modulation and thickness modulation

According to the generalized Meiklejohn-Bean model, $H_{EB}$ can be modulated by both the interlayer spacing of the AFM/FM heterostructure, and the thickness of AFM or FM material. Thickness modulation represents the traditional device performance modulation method by changing the material thickness. Since the thickness cannot be modulated in situ, so new devices need to be prepared. LS modulation represents a novel approach to device performance modulation by changing the interface spacing. LS modulation can be modulated in situ directly on the fabricated devices, showing a more convenient application potential. Equation (1) shows that $H_{EB}$ increases to saturation with the increase of $t_{AFM}$ and decreases with the increase of $t_{FM}$, therefore, the effect of FM or AFM thickness on $H_{EB}$ for the FPSe/FGT heterostructure was investigated. The anisotropy energy that can pin the FM spins during magnetization reversal, is enhanced upon increasing the AFM thickness, and $H_{EB}$ is also strengthened concomitantly. Figure 3a shows the dependences of AFM thickness $t_{FPSe}$ on $H_{EB}$, where $H_{EB}$ increases from 28 mT (FPSe (17.2 nm)/FGT (26.9 nm)) to 62 mT (FPSe (21.6 nm)/FGT (26.9 nm)). To further explore the effect of

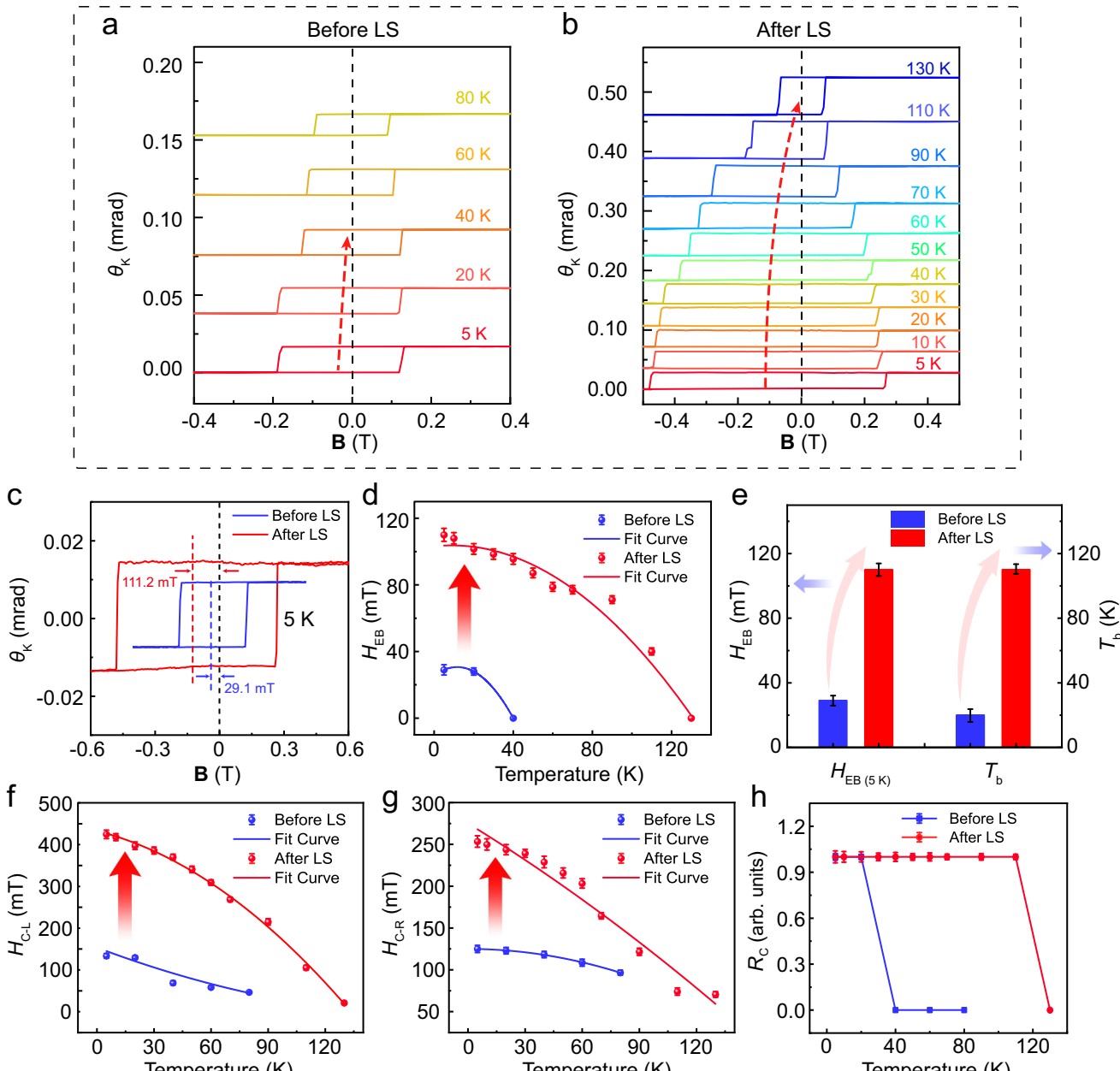

**Fig. 2 | Enhanced EB effect through LS. a, b** The temperature-dependent Kerr loops of FPSe/FGT heterostructures before and after LS. **c** Kerr loops versus the magnetic field for FPSe/FGT heterostructure through the magneto-optical Kerr test before and after LS at 5 K. **d** A detailed comparative investigation for $H_{EB}$ and $T_b$ versus the temperature before and after LS at 5 K. **e** Summary of $H_{EB}$ and $T_b$ enhancements before and after LS. **f, g** A detailed comparative investigation for $H_{C-L}$ (**f**) and $H_{C-R}$ (**g**) versus the temperature before and after LS at 5 K. **h** $R_C$ versus the temperature for the FPSe/FGT heterostructure before and after LS. Error bars in **d**, **e**, **f**, **g**, and **h** represent standard deviation for several consecutive measurements.

$t_{FPSe}$ on $H_{EB}$, more FPSe/FGT heterostructures (FPSe with different thicknesses, and FGT with a certain thickness of ~18.0 nm) were prepared and measured in Supplementary Fig. 13a. In addition, we also investigated the dependences of $H_{EB}$ on the thickness of FM ($t_{FGT}$). Figure 3b displays the hysteresis loops for FPSe (~27.0 nm)/FGT (18.0 nm, 22.4 nm, and 25.5 nm, respectively) heterostructures at 5 K, where $H_{EB}$ decreases from 32 mT (FPSe/FGT, 27.0 nm/18.0 nm) to 10 mT (FPSe/FGT, 27.0 nm/25.5 nm). The dependences of $H_{EB}$ on $t_{FGT}$ are summarized in Supplementary Fig. 13b, showing that $H_{EB}$ decreases upon increasing $t_{FGT}$, which satisfies the FM-thickness dependence of $H_{EB}$ in conventional exchange-biased systems. In contrast, interlayer spacing modulation becomes a more stable and efficient modulation method with a lower cost. According to the above experimental results, the weak EB effect in FPSe/FGT heterostructure is extremely sensitive

to its interlayer spacing. We have investigated the effect of various out-of-plane pressures on the EB effect, and temperature-dependent Kerr loops of FPSe/FGT heterostructures (sample B, optical image in Supplementary Fig. 14) pressed under 0, -8, -11, and -13 GPa at 5 K were measured, respectively (Fig. 3c). Under 0 GPa, a slightly shifted loop signifies weak magnetic interface coupling in FPSe/FGT heterostructures. As the pressure increases, $H_{EB}$ enhances. $T_b$ is also sensitive to pressure, increasing from 20 to 110 K which is close to the Néel temperature, when the pressure varies from 0 GPa to -13 GPa. The coercive field ($H_C$) is also remarkably enhanced as the pressure increases, as shown in Supplementary Fig. 15. The performance comparison before and after LS is shown in Supplementary Table 5. Figure 3d, e illustrates the evolution of $H_{EB}$ in thickness-modulation and interlayer spacing modulation. Furthermore, we compared $H_{EB(20\ K)}$

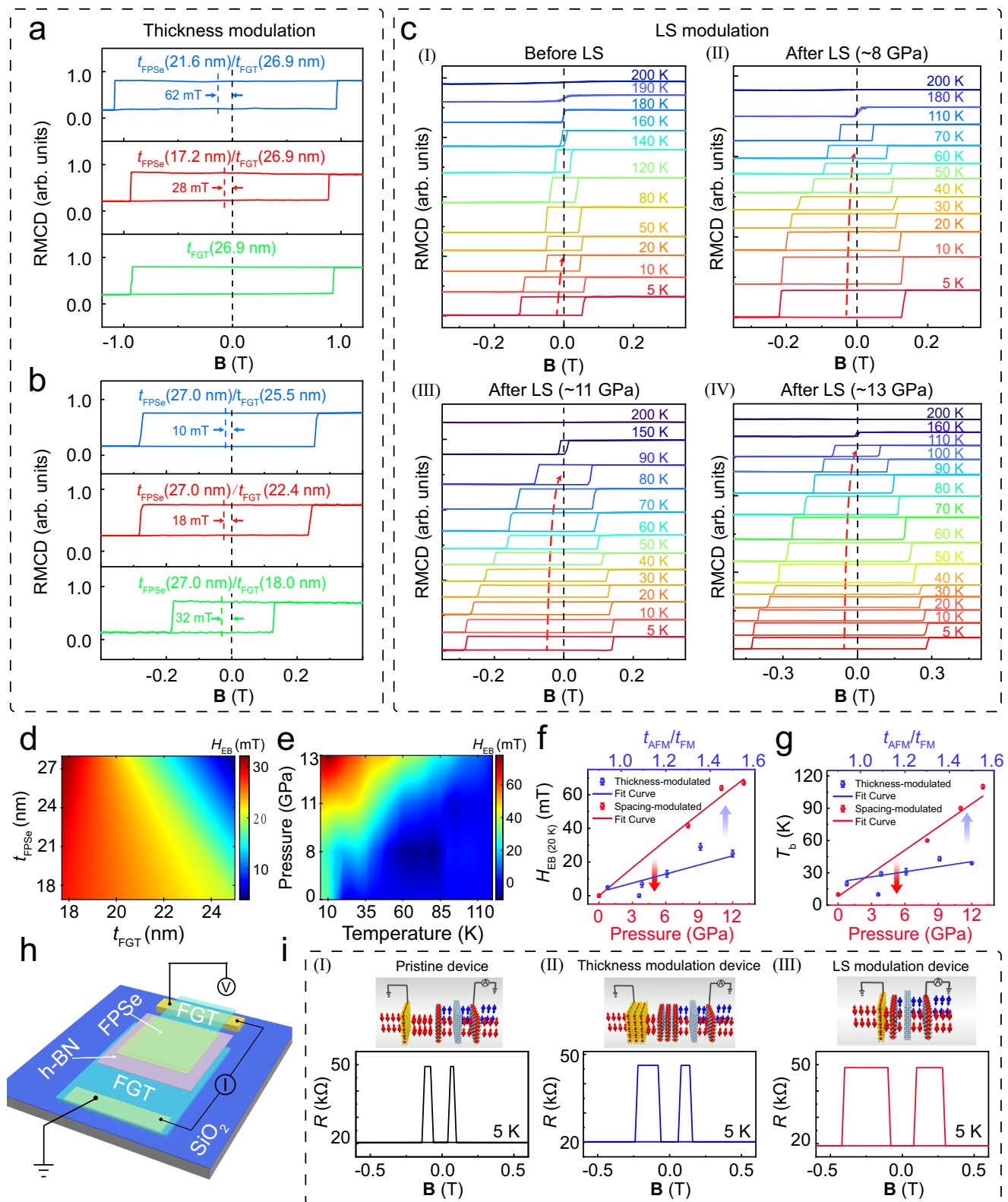

**Fig. 3 | Comparison of interlayer spacing-modulation and thickness-modulation. a** Kerr loops for individual FGT (26.9 nm) and FPSe (17.2 nm, 21.6 nm)/FGT (26.9 nm) heterostructures, respectively, measured at 5 K. **b** Kerr loops for individual FPSe (27.0 nm)/FGT (18.0 nm, 25.5 nm, and 22.4 nm) heterostructures, respectively, measured at 5 K. **c** The temperature-dependent Kerr loops of FPSe/FGT heterostructures are measured under 0 Gpa, -8 Gpa, -11 Gpa, and -13 Gpa at 5 K, respectively. **d** The evolution of $H_{EB}$ on $t_{FGT}$ and $t_{FPSe}$. **e** The evolution of $H_{EB}$ on temperature and pressure before and after LS. **f, g** The evolution of $H_{EB}$ and $T_b$ on pressure and $t_{AFM}/t_{FM}$. Error bars represent standard deviation for several consecutive measurements. **h** Schematic diagram of spin valve device. **i** Tunneling resistance of the pristine device (I), thickness modulation device (II), and LS modulation device (III) with B applied parallel to the FGT $c$-axis. Device structure diagrams are shown in the upper part. Thickness modulation is achieved by constructing a new heterojunction. pristine device and thickness modulation device have different top FGT and FPSe thicknesses. LS modulation reduces the interlayer spacing of pristine device by applying pressure in situ.

and $T_b$ as a function of pressure and $t_{AFM}/t_{FM}$, as shown in Fig. 3f, g. $H_{EB(20\ K)}$ is defined as the exchange bias field at 20 K. $H_{EB(20\ K)}$ and $T_b$ show a linear correlation with the increase of pressure by fitting. Compared to thickness-modulation, the interlayer spacing-modulation results reveal a clear pressure-induced magnetic coupling enhancement, offering important insights into the understanding of the physical nature of the interlayer magnetic states.

Based on the above experimental results, we fabricated high-quality tunneling spin valves (FPSe/FGT/h-BN/FGT, with optical image and atomic force microscope image illustrated in Supplementary Figs. 16 and 17) using the dry transfer technique in a glove box to investigate the spin valve behavior under various modulation methods. Figure 3h shows a schematic diagram of the spin valve device. The pristine spin valve device and the spin valve device modulated by two modulation modes are shown in the upper schematic diagram of Fig. 3i. To demonstrate spin valve behavior, the resistance ($R$) of the MTJ as a function of a perpendicular magnetic field was measured. Figure 3i shows the results measured at 5 K. For the pristine device (the optical image in Supplementary Fig. 16), as B was swept from negative to positive values, the resistance suddenly increased from approximately ~20 to ~49 kΩ at ~60 mT, followed by a sudden decrease back to ~20 kΩ at ~80 mT. As the magnetic field was swept back, an analogous abrupt increase and equally abrupt decrease in tunneling magnetoresistance were observed at ~−80 mT and ~−120 mT. This was precisely the behavior expected for a tunneling spin-valve due to the hysteretic magnetization switching of two ferromagnetic electrodes. For thickness modulation, a new device (thickness modulation device, the optical image in Supplementary Fig. 17) was constructed to investigate thickness modulation since thickness cannot be modulated in situ, whose thicknesses of top FPSe and FGT were different from that of the pristine device. For LS modulation, LS was processed directly on the pristine device to modulate the performance. Spin valve behavior was observed after both modulations (Fig. 3i). Supplementary Fig. 18 shows the resistance of the MTJ as a function of magnetic field (B) from 5 to 120 K. The magnitude of the TMR was defined as $(R_{AP} − R_P)/R_P$, which characterizes the transmission efficiency, where $R_{AP}$ and $R_P$ are the resistance obtained for parallel and antiparallel alignments of the magnetization. The TMR values were 141%, 130%, and 154% at 5 K for the pristine device Fig. 3i (I), thickness modulation device (Fig. 3i (II)), and LS modulation device (Fig. 3i (III)), respectively. The maximum field window of the thickness modulation device and LS modulation device is 3.5 and 7.5 times larger than that of the pristine device, respectively. Compared to thickness modulation, interlayer spacing modulation is more stable and efficient with a reduced cost. To further verify the significant performance enhancement after LS modulation, the tunnel resistance of thickness modulation device was measured after LS, showing a three-fold enhancement (Supplementary Fig. 19). In addition, the TMR and the field window of the FGT/h-BN/FGT spin valve before LS was 119% and 20 mT, respectively, as shown in Supplementary Fig. 20. Compared to FGT/h-BN/FGT spin valve in our work, the field window under LS modulation was around 15 times larger than that of the vertical FGT/h-BN/FGT spin valves at 5 K. The experimental results confirmed the great advantage of LS modulation in realizing high-performance spin valve devices.

**Non-local EB effect in magnetic vdW-structure**
Generally, the hysteresis loops of FGT flake remain symmetric without any magnetic EB, unless capped with FPSe[18,19]. Nevertheless, the $H_{EB}$ of FGT is also observed at point 2 (Fig. 4a, sample B), a phenomenon that has not been reported yet. Unexpected magnetic hysteresis behavior is observed in the exposed FGT region connected to the heterostructure (position 2), even without LS (Fig. 4b). A 27 mT $H_{EB}$ and a 20 K $T_b$ are observed as shown in Fig. 4b (II), consistent with FPSe/FGT (position 1). Importantly, the measured $H_{EB}$, $T_b$, and $H_C$ of this connected FGT (position 2) exhibit basically identical reinforcement after LS, as shown

in Fig. 4c (I) and 4c(II). However, we have also measured the hysteresis loops of isolated FGT (position 3) before and after LS to exclude pressure-induced magnetic hysteresis behavior of FGT, and the results show a lack of the EB effect (position 3) either with or without LS (Fig. 4b (III) and 4c(III)). In addition, to prove that the phenomenon is prevalent, we also fabricated other FPSe/FGT heterostructures containing both connected and isolated FGT (sample C, see Supplementary Information). As shown in Supplementary Fig. 21, the hysteresis loops of isolated FGT remain symmetric without any magnetic EB, while the connected FGT and FPSe/FGT heterostructure exhibit similar $H_{EB}$, $T_b$, and $H_c$. Figure 4d shows the comparison of $H_{EB}$ and $T_b$ at three positions. Before LS, positions 1 and 2 show the same EB effect, but the EB effect is absent at position 3. After LS, $H_{EB}$, $T_b$, and $H_C$ show consistent enhancement at positions 1 and 2, and there is still no EB effect at position 3. The temperature-dependent scatter plots of the left and right coercive fields at all three positions are shown in Fig. 4e. The difference between $H_{EB}$ gradually decreases from 71 mT and 68 mT (at 5 K) to 0 mT (at 90 K) for positions 1 and 2, respectively, which are summarized in Fig. 4f. This result indicates that the modulation of $H_{EB}$ and $T_b$ at positions 1 and 2 are basically the same. This non-local coupling mechanism of the horizontal pinning originates from the inherent influence of the FGT. As long as the FGT is connected with FPSe, the scale of transverse propagation can reach tens of micrometers or even hundreds of micrometers, leading to a large horizontal pinning. Due to the limitation of mechanically peeled materials, the exact boundary of the transverse propagation effect has not been studied further. This wide range of EB effect and propagation effects at the micrometer level will be beneficial for the application of industrial devices.

To investigate the non-local coupling mechanism of the horizontal pinning, we propose a model based on the experimental results above, which suggests the possible spin structures in the inset of Fig. 4g, h. First, we observed this non-local effect below the Néel temperature of 110 K. Fei et al. pointed out that FGT exhibits single-domain properties at temperatures below 150 K when its thickness falls between 10 and 100 nm[38]. The single-domain state of FGT was proved in new FPSe/FGT heterostructures (Supplementary Fig. 22a), with the MOKE mappings of the FGT and heterostructure measured at 5 K (Supplementary Fig. 22e–g). The magnetism in areas of different thicknesses within one flake was flipped simultaneously, clearly indicating the single-domain nature of FGT flakes and confirming the sample homogeneity. Furthermore, we measured the non-local exchange bias field in two heterostructures with various thicknesses (sample D and sample E) (Supplementary Figs. 22 and23). By comparing the RMCD signals of the heterostructure regions with those of the adjacent connected bare FGT region, the non-local exchange bias field across the same sample was homogeneous. Therefore, within the range of temperature and sample thickness measured in this experiment, FGT can be considered as the single-domain configuration. When FGT was covered by FPSe, the spin direction of FGT of the heterostructure was pinned by the antiferromagnetic material. Due to the single-domain nature of FGT, the spin of connected FGT was pinned in the same manner, leading to the non-local EB effect. As shown in Fig. 4g, h, when no pressure was applied, the pinning effect is not obvious due to the weak interface coupling, corresponding to a small $H_{EB}$ and a low $T_b$ experimentally. When sufficient pressure was applied, the enhanced interlayer coupling led to an increase of the pinning effect, which caused larger coercive fields in the hysteresis loop, as well as larger $H_{EB}$ and higher $T_b$ in the experiments. Moreover, this distance at which EB can be detected in connected FGT can go beyond 40 μm in our samples after LS (Supplementary Fig. 21, where the maximum distance is limited by the length of the sample), showing a simultaneous enhancement between FPSe/FGT heterostructures and connected FGT. This phenomenon can provide a potential platform for studying the experimental determination of spin transport properties. Moreover, understanding of the non-local coupling mechanism of the

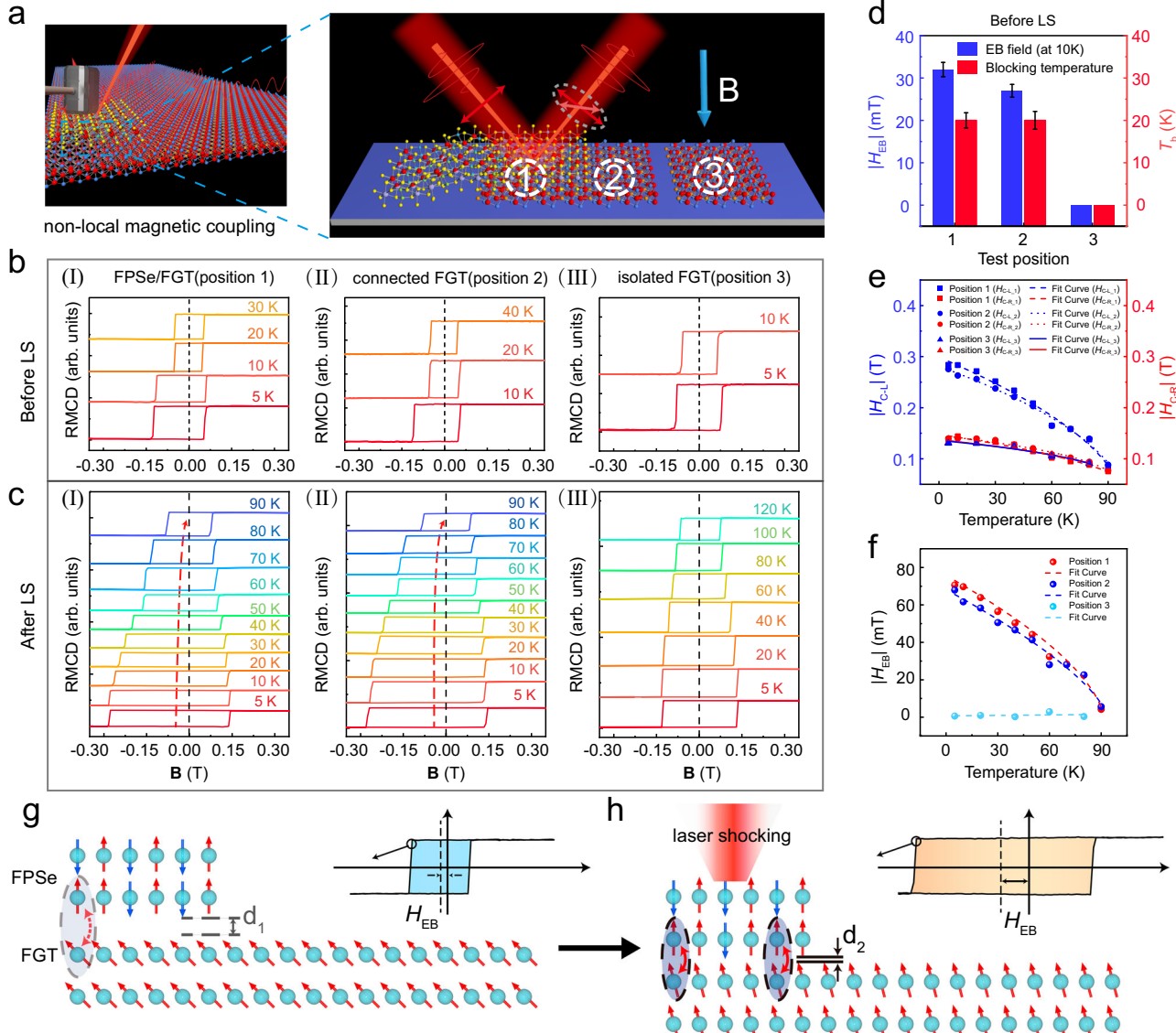

**Fig. 4 | Non-local effects of EB. a** The schematic diagram of the magneto-optical Kerr test for sample B, where position 1, 2, and 3 represents FGT/FPSe, connected FGT, and isolated FGT. **b** The measured EB field for FPSe/FGT at position 1 (I), connected FGT at position 2 (II), and isolated FGT at position 3 (III) before LS, respectively. **c** The measured EB field for FPSe/FGT at position 1 (I), connected FGT at position 2 (II), and isolated FGT at position 3 (III) after LS, respectively. **d** A comparison of $H_{EB}$ and $T_b$ at the three positions corresponds to (**b**). Error bars represent standard deviation for several consecutive measurements. **e** A detailed comparison of $H_{C-L}$ and $H_{C-R}$ versus the temperature of FPSe/FGT heterostructures, connected FGT, and isolated FGT under -11 GPa corresponds to (**c**). $H_{C-L\_1}$, $H_{C-L\_2}$, and $H_{C-L\_3}$ represent a left coercive field, $H_{C-L}$, at position 1, position 2, and position 3, respectively. $H_{C-R\_1}$, $H_{C-R\_2}$, and $H_{C-R\_3}$ represent a right coercive field, $H_{C-R}$, at position 1, position 2, and position 3, respectively. **f** A detailed comparative investigation for $H_{EB}$ versus the temperature of FPSe/FGT heterostructures, connected FGT, and isolated FGT under -11 GPa corresponds to (**c**). Error bars in **e** and **f** represent standard deviation for several consecutive measurements. **g**, **h** Schematic diagrams of a model for the spin structure between the FPSe/FGT heterostructure and connected FGT ($d_1 > d_2$).

horizontal pinning will ultimately contribute to the development of the spin logic device adopting 2D FM as a spin transport channel.

## Discussion and prospect

Devices based on 2D MTJs and 2D spin valves are becoming the most promising candidates for future development of MRAM and magnetic sensor applications[51,52]. The reliability of the device is becoming an increasingly important technical issue. A large EB effect between FM and AFM promotes magnetic stability of the pinned layer FM, which is key to producing a reliable, high-quality spintronic device with improved resistivity to magnetic noise. Our work has demonstrated the existence of EB effects in vdW magnetic heterostructures and achieved great enhancement of $H_{EB}$, $T_b$, and $H_C$ through vdW spacing tuning. It is worthwhile to emphasize that the interlayer coupling of

vdW heterostructures can be manipulated promptly over a large area via LS. This provides the potential to promote MRAM with more stable performance and higher storage density (Supplementary Fig. 24). This method can be extended to other different types of vdW heterostructures. Here, the EB effect can greatly enhance the stability and scalability of the magnetic device. The use of the manipulation method and the selection of suitable magnetic materials hold the potential of realizing 2D vdW heterostructures memory devices at room temperature.

On the other hand, the EB effect shown in connected FGT is partially consistent with the phenomenon observed in heterostructures. This non-local effect of $H_{EB}$ is enhanced after LS. According to the current experimental data, $H_{EB}$ can propagate over 40 μm without degradation. Unfortunately, limited by the experimental conditions,

we are unable to measure the maximum extent that this lateral EB effect can reach. It is expected to realize the transmission of more than 100 μm or even hundreds of micrometers. This effect can enhance the function of spintronic devices and expand their design flexibility. Investigation into its mechanism is also an essential task in the future.

With the above effects, novel functional devices with a much simpler structure can be designed. For example, by means of the EB effect with enhanced interlayer coupling in vdW heterostructures, a larger storage window can be obtained in a spin valve or an MTJ. Moreover, the $H_{EB}$ of a specific device can be effectively regulated by the external pressure field without the need to re-manufacture the device, compared to the traditional adjustment of the $H_{EB}$ field by changing the AFM and FM layer thicknesses. On the other hand, the non-local effect may even enable the magnetic property regulation in suspended 2D materials, which has been extremely difficult. Further-more, the non-local effect can be utilized to resolve a long-lasting problem in nanoscale MTJs. Traditionally, the free layer and the fixed layer in an MTJ need to be perfectly aligned, proposing a great chal-lenge for the photolithography process. However, using non-local effects, the two layers of FM material do not have to be perfectly aligned, which could greatly facilitate the manufacturing process.

In conclusion, we have successfully established a strong magnetic coupling that yielded a stable and considerable exchange bias effect, by achieving the tuning of vdW spacing in 2D magnetic FPSe/FGT heterostructures through pressure engineering. Then, we demon-strate a spin valve device with high sensitivity and robustness. Fur-thermore, we also observe the exchange bias effect in the connected FGT, which is consistent with the regulation of the coupling strength of the FPSe/FGT heterostructures. Our findings provide novel methods for the manufacture, exploration, and tuning of magnetic vdW het-erostructures with strong interlayer coupling, opening the possibility for customized 2D spintronic devices in the future, such as high-density magnetic storage, and more possible applications in other fields including logic devices.

## Methods

### Details of laser shock processing

The FPSe/FGT vdW heterostructures placed on $SiO_2$ (300 nm)/Si substrate were nano-strained by employing laser shocking pressure. Figure 1bII shows a schematic of the laser shocking process. First, A $SiO_2$ (300 nm)/Si substrate with FPSe/FGT heterostructure was placed on a slide and covered with a clean metal film. Here, the metal film was a 10 μm aluminum (Al) foil, which functioned as a buffer layer and a contaminant barrier due to its excellent ductility and low cost. Further, nano-graphene powder (Hangdan Optoelectronics Technology, China) was uniformly spun onto Al foil as an ablative coating with a thickness of approximately 1.0 mm. Note that the range of the spin-coated graphene powder needs to completely cover the $SiO_2$ (300 nm)/Si substrate area. A cover glass was then placed over the graphene powder to serve as a shock confinement layer. In this way, the FPSe/FGT heterostructure was successfully sandwiched between two glass sheets. Finally, multiple laser pulses were implemented by placing the prepared sample on an X–Y–Z computer-controlled motorized stage. The laser position was fixed by adjusting the laser focus. And the step size and path of the X–Y–Z electric platform were set to realize the pressure application in the customizable area.

Next, a pulsed (10 ps) Q-switch Nd:YAG laser (Continuum Sur-elite III, Wavelength: 1064 nm, Power: 0.3–1.4 GW/cm$^2$) was used as an energy source for ablation. The laser beam diameter of 4 mm was attained by a focusing lens, which was calibrated by a photo-sensitive paper (Kodak Linagraph, type: 1895). The laser pulse irradiated the ablative coating (1.0 mm graphene powder). Shock waves were generated by sublimation of the ablative layer using laser pulses. Due to the constraints of the confinement layer, the

shock wave could propagate through the Al foil and apply pressure at the nanoscale resolution on the $SiO_2$ (300 nm)/Si substrate with FPSe/FGT heterostructure. The pressure depends on the intensity of the laser and the duration of the pulse during the laser shocking process.

### Fabrication of FPSe/FGT heterostructures

The FPSe/FGT vdW heterostructures presented in this work were fabricated by a common dry transfer method utilizing a poly(-dimethylsiloxane) (PDMS) carrier in a glove box. First of all, thin FGT flakes were exfoliated from bulk crystal using Scotch tape and then transferred onto $SiO_2$/Si substrates. Next, thin FPSe flakes were exfo-liated onto the PDMS carrier from an as-grown FPSe crystal and then directly transferred onto the FGT flake under the optical microscope assisted by an aligned transfer system. Other vdW heterostructures were prepared with a similar method.

### Magneto-optic Kerr effect (MOKE) spectroscopy

The polar MOKE measurements were performed using a standard lab-made MOKE setup. A 633 nm polarized laser with the intensity of 3 μW was used as a light source. The temperature range of the setup was 4.2–300 K, and the magnetic field range was 0–5 T. During the mea-surement, a chopper and a photoelastic modulator were used to improve the signal-to-noise ratio of the Kerr signal and the laser was irradiated normally on to the surface of the sample to detect the out-of-plane magnetism. The FPSe/FGT heterostructure went through a cooling step from 160 K (a temperature between the Néel temperature of FPSe and the Curie temperature of FGT) down to the lowest tem-perature 5 K, with a large cooling field of 1 T (the cooling field was much larger than the coercive field of FGT).

### Characterizations of heterostructures

For FPSe/FGT vdW heterostructures, optical imaging and mor-phology were acquired by optical microscope and atomic force microscope, respectively. Raman measurements were performed using Laser Confocal Raman Spectrometer with a laser wavelength of 532 nm (LabRAM HR800, Horiba JobinYvon, France). The SEM equipped with an EDS were performed to check the quality, orientation, topography, and composition of the FGT and FPSe single crystals, respectively.

### Measurements of spin valves

Hall effect measurements were performed in a Quantum Design phy-sical property measurement system (PPMS) with the temperature ranging from 300 to 2.5 K. The magnetic field was applied normal to the 2D plane during all the measurements.

## Data availability

All data that support the findings of this study are available in the Figshare database at the following https://doi.org/10.6084/m9.figshare.22372912.

## Code availability

The codes used in this paper are available from the corresponding author upon reasonable request.

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

## Acknowledgements

This work was supported by the National Natural Science Foundation of China (grants 62222404, 61974050, 11704138, 11404124), National Key Research and Development Plan of China (grants 2019YFB2205100, 2021YFB3601200), Research Grants Council of Hong Kong (Grant No.

PDFS2223-4SO6), Innovation Fund of the Wuhan National Laboratory for Optoelectronics, Open Fund of State Key Laboratory of Infrared Physics. J.H. thanks the financial and technical supporting of magneto-optical measurement station in Wuhan National High Magnetic Field Center.

## Author contributions

L.Y. and J.H. supervised the project. H.D. and H.C. synthesized and characterized the bulk Fe3GeTe2 and FePSe3 crystals. X.H. fabricated the samples and analyzed the vdW gaps in FPSe/FGT heterostructures, assited by L.T. L.Z. performed the MOKE and RMCD mesurements with the assistance of H.D. and H.C. G.C. and C.B. provided simulations; G.C. peformed laser shocking; X.H. and L.Z. analyzed the data; X.H., L.Z., J.X., J.H., K.X., and L.Y. prepared the paper with input from all authors; Z.L., Z.P., R.L., and W.S. provided many critical comments and analyses.

## Competing interests

The authors declare no competing interests.
