## [Peer Review File · Nature Communications]

Reviewers' Comments:

Reviewer #1:

Remarks to the Author:

This is an extremely interesting paper and it will have a very high impact in the field vdW heterostructures. Using the laser shocking technique, the authors realized largely enhanced EB effect and spin valve devices. The pronounced enhancement of EB and coercivity demonstrates the largely increased interlayer coupling. The technique is a standard technique used in the research of magnetic recording and can be easily extended to other research fields based on vdW heterostructures. I believe that many researchers in the community will be very interested in this paper. I strongly recommend the publication in Nature Communications. The paper is well written. The authors only need to tackle the issue of references as shown below.

The authors may ignore two recent papers about electrically manipulation of the interlayer coupling in vdW heterostructure and EB. I. 'Electric Control of Exchange Bias Effect in FePS₃-Fe₅GeTe₂ van der Waals Heterostructures', *Nano Lett.*, 22, 6166 (2022). II. 'Gate-tuned interlayer coupling in van der Waals ferromagnet Fe₃GeTe₂ nanoflakes', *Phys. Rev. Lett.* 125, 047202 (2020).

Reviewer #2:

Remarks to the Author:

Comments on "Manipulating Exchange Bias in 2D Magnetic Heterojunction for High-performance Robust Memory Applications":

I have carefully read the manuscript by X. Huang et al. which studies exchange bias in Fe₃GeTe₂ (FGT) / FePSe₃ heterostructures exposed to intense laser shock processing. Notably, the authors report an enhancement in the exchange bias field which they attribute to an enhancement of the interlayer coupling due to irreversible changes to the van der Waals interlayer spacing. I found the work interesting, but there are a number of key issues which must be addressed before any decision can be made.

- 1) The integrity of the FGT after the laser shock process must be confirmed, especially since exchange bias fields of similar magnitude have been reported in oxidized Fe₃GeTe₂ alone (H. K. Gweon, et al. *Nano Letters* 2021). In the present work, the STEM image of the FGT before and after laser shock (Fig. 1e and 1f) looks very different. This important change is seemingly never addressed in the text. Can the authors explain its origin?
- 2) The authors provide Raman data of the flakes and heterostructure before laser shock. Can they provide Raman for the flakes and heterostructure after the processing to confirm there is no damage?
- 3) The FGT in sample B, which is used to study the non-local exchange bias effect clearly has large variations in sample thickness (Fig. S13). In fact, it looks like the flake contains both very thick and very thin regions. Considering the authors also establish that the exchange bias field is strongly dependent on FGT thickness, I am confused by how the entire flake can switch at the same time. The authors should provide spatial dependent MOKE maps to understand how the varying thickness can effect the non-local exchange bias effect and the sample homogeneity.
- 4) Since the exchange bias should be an interfacial effect, it is not clear why the FePSe₃ thickness should matter as long as it remains in the thin bulk limit. This seems to be confirmed by Supplementary Figure 12, and the lack of any tAFM term in the Mauri model. Yet, the exchange bias in Fig. 3b more than doubles with a change of only 4 nm (from 17.2 nm to 21.6 nm). Can the authors explain this discrepancy?
- 5) FePSe₃ is a zig-zag AFM. Do the authors expect that there could be a spatially varying spin texture induced in the FGT as a result?

In addition to the above scientific questions, I have a number of comments which may improve the readability and presentation of the manuscript:

1) Fig. 1a i and ii are never explained or mentioned anywhere in the text. In my opinion, this panel is not very useful to the story and could be removed entirely. Similarly, Fig. 1b i-ii are not directly addressed in the text, and Fig. 1b ii gives the impression that the interlayer spacing can be modulated back and forth (modulation and unmodulation). From my understanding, the laser shock pressure is irreversible, therefore this panel is misleading.

2) I found that several figure panels were repetitive, redundant, or unnecessary. For instance, Fig. 1c, 2a, and 3a are very similar, and Supplementary figure 7 seems unnecessary.

3) The authors notation of States 1-3 for the spin valve device was confusing and the bottom panel in Fig. 3i is not useful. It would be clearer to just use "Pristine", "Thickness Modulated", and "Laser Shock Modulated". In fact, it is not clear what the thickness modulation refers to since the thickness cannot be modulated in situ. If it refers to separate devices, the details should be clearly presented. The same is true for comparison between all the different devices. The authors claim significant enhancements, but the paper needs a clearer comparison of the effect the various FGT and FePSe3 thickness can have on the individual device performance.

4) Since laser shockwave experiments are relatively unexplored so far in the field of 2D materials, further details and depictions of the experimental setup would be greatly appreciated and would make the paper much more understandable for a general audience. The methods section should therefore be greatly expanded.

Response Letter

We sincerely thank you and the reviewers for the great efforts providing valuable comments concerning our manuscript entitled “Manipulating Exchange Bias in 2D Magnetic Heterojunction for High-performance Robust Memory Applications” (ID: NCOMMS-22-35123). These comments are all helpful for revising and improving our paper. We have fully addressed the comments carefully and revised the manuscript. The following is the point-to-point response for the comments.

Reviewer #1

This is an extremely interesting paper and it will have a very high impact in the field vdW heterostructures. Using the laser shocking technique, the authors realized largely enhanced EB effect and spin valve devices. The pronounced enhancement of EB and coercivity demonstrates the largely increased interlayer coupling. The technique is a standard technique used in the research of magnetic recording and can be easily extended to other research fields based on vdW heterostructures. I believe that many researchers in the community will be very interested in this paper. I strongly recommend the publication in Nature Communications. The paper is well written. The authors only need to tackle the issue of references as shown below.

Response: We are grateful for the reviewer’s comments and appreciation for our work.

1. The authors may ignore two recent papers about electrically manipulation of the interlayer coupling in vdW heterostructure and EB. I. Electric Control of Exchange Bias Effect in FePS₃–Fe₅GeTe₂ van der Waals Heterostructures’, Nano Lett, 22, 6166 (2022). II. ‘Gate-tuned interlayer coupling in van der Waals ferromagnet Fe₃GeTe₂ nanoflakes’, Phys. Rev. Lett. 125, 047202 (2020).

Response: We appreciate the reviewer’s valuable comments. We have paid careful attention to the above two works, the contents of which provide important guidance and

reference for our work. We have cited these two references in the revised manuscript.

The revised contents in the manuscript are listed below:

(1) Page 3, lines 50-52:

Recently, the EB effect of magnetic vdW heterostructures has been studied in $\text{CrCl}_3/\text{Fe}_3\text{GeTe}_2^{23}$, $\text{MnPS}_3/\text{Fe}_3\text{GeTe}_2^{24}$, $\text{MPSe}_3/\text{Fe}_3\text{GeTe}_2^{25}$, $\text{FePS}_3\text{-Fe}_5\text{GeTe}_2^{26}$, and $\text{Fe}_3\text{GeTe}_2^{27}$ systems.

References

- 26 Albarakati, S. et al. Electric Control of Exchange Bias Effect in $\text{FePS}_3\text{-Fe}_5\text{GeTe}_2$ van der Waals Heterostructures. *Nano Lett.* **22**, 6166-6172 (2022).
- 27 Zheng, G. et al. Gate-Tuned Interlayer Coupling in van der Waals Ferromagnet Fe_3GeTe_2 Nanoflakes. *Phys. Rev. Lett.* **125**, 047202 (2020).

Reviewer #2

I have carefully read the manuscript by X. Huang et al. which studies exchange bias in Fe_3GeTe_2 (FGT) / FePSe_3 heterostructures exposed to intense laser shock processing. Notably, the authors report an enhancement in the exchange bias field which they attribute to an enhancement of the interlayer coupling due to irreversible changes to the van der Waals interlayer spacing. I found the work interesting, but there are a number of key issues which must be addressed before any decision can be made.

Response: We sincerely thank the reviewer for the great efforts providing valuable suggestions.

1. The integrity of the FGT after the laser shock process must be confirmed, especially since exchange bias fields of similar magnitude have been reported in oxidized Fe_3GeTe_2 alone (H. K. Gweon, et al. Nano Letters 2021). In the present work, the STEM image of the FGT before and after laser shock (Fig. 1e and 1f) looks very different. This important change is seemingly never addressed in the text. Can the authors explain its origin?

Response: We gratefully appreciate the reviewer's valuable comments. We have performed new experiments of the Raman spectrum and mappings of FGT before and after laser shocking as shown in the revised Supplementary Fig. 8, to reveal the integrity issue. According to the results, there is a negligible change, indicating that FGT sustained its good quality.

The preparation and testing processes in our work were carried out in the glove box and vacuum chamber with only the laser shocking processing being exposed to air within minutes to avoid oxidation. In addition, according to the MOKE results of the FGT after laser shocking (Supplementary Fig. 11 and 21 in the Supplementary Information), the exchange bias field was absent, excluding FGT oxidation as the origin of the exchange bias field.

Fig. 1e and 1f show STEM images along different cross-sectional directions of the same sample. The cross-sectional orientations are difficult to keep in situ completely because we must remove the sample outside the chamber to conduct laser shocking. Despite rotating the cross-sectional direction, the lattice structure of FGT is consistent with that of the corresponding crystallographic direction. This indicates that the reduced interlayer distance after laser shocking without damaging the lattice structure. We have revised our claims more clearly in the revised manuscript, as highlighted in blue and shown below.

The revised contents:

(1) Page 7, lines 125-133:

Further, we used cross-sectional (scanning transmission electron microscopy) STEM images to measure the interface gap width of the FPSe/FGT heterostructure. The STEM images of the samples before and after LS were acquired along different orientations because the laser shocking process outside the chamber made it hard to conduct in-situ characterizations. After LS, vdW interlayer distances of FPSe and FGT were fixed at their intrinsic values of ~ 2.5 Å and ~ 2.1 Å respectively, consistent with previous reports^{40, 45-47}, and there was no damage to the lattice structure, but the interface gap width (initial ~ 7.4 Å) was dramatically reduced to ~ 4.6 Å (Fig. 1d, e).

Supplementary Fig. 8 | Characterization of FGT after LS. **a, b** The optical image and the SEM image of FGT after LS. **c** Raman spectrum of FGT before and after LS. **d, e** Raman mapping of E_{2g}^2 peak intensity of FGT flake before (**d**) and after (**e**) LS. The corresponding scanning areas before and after LS were marked by the blue and red boxes in (a) respectively.

2. The authors provide Raman data of the flakes and heterostructure before laser shock. Can they provide Raman for the flakes and heterostructure after the processing to confirm there is no damage?

Response: We gratefully appreciate the reviewer's valuable comments. We have added new Raman characterizations for the flakes and heterostructure after laser shocking processing in the revised Supplementary Fig. 9, as shown below. The mapping results show that the positions and intensities of the characteristic peak E_{2g}^1 (90 cm^{-1}) of FGT and characteristic peak A_{1g} (215 cm^{-1}) of FPSe remain consistent before and after laser shocking, showing no damage to the samples after laser shocking processing and indicating the integrity of the flakes and heterostructure.

Supplementary Fig. 9 | Characterization of FPSe and FPSe/FGT heterostructure after LS. **a, b** The optical image and the SEM image of FPSe/FGT heterostructure after LS. **c, d** Raman spectrum of FPSe and FPSe/FGT heterostructure before and after LS. In Fig. 2d, P₁, P₂, and P₃ marked in green in (d) represent the characteristic peak of FPSe flake in (c). P₁, P₂, and P₃ marked in blue represent the characteristic peaks E_{2g}¹, E_{2g}², and A_{1g}¹ of FGT flake. **e-h** Raman mapping of FPSe/FGT heterostructure before (**e, f**) and after (**g, h**) LS. The blue mappings represent E_{2g}¹ peak (~90 cm⁻¹) intensity of FGT flake and the green mappings represent A_{1g}¹ peak (~215 cm⁻¹) intensity of FPSe flake. The areas marked by the blue and red boxes in (a) represent the test ranges before and after LS respectively.

3. The FGT in sample B, which is used to study the non-local exchange bias effect clearly has large variations in sample thickness (Fig. S13). In fact, it looks like the flake contains both very thick and very thin regions. Considering the authors also establish that the exchange bias field is strongly dependent on FGT thickness, I am confused by how the entire flake can switch at the same time. The authors should provide spatial dependent MOKE maps to understand how the varying thickness can effect the non-local exchange bias effect and the sample homogeneity.

Response: Thanks for the reviewer's valuable comments. We obtained the thickness dependence of the local exchange bias effect by comparing different samples while investigated the synchronous non-local exchange bias effect within the same flake. FGT

exhibited a homogeneous single-domain configuration within the same flake, regardless of the thickness distribution, as explained in the manuscript¹⁻³. Therefore, the same piece of FGT can be magnetized at the same time to form a homogeneous non-local exchange bias field and switch synchronously, independent of thickness.

We have performed new experiments to show this thickness independence of the non-local exchange bias effect and the sample homogeneity within the same flake, which was proved in new FPSe/FGT heterostructures (Supplementary Fig. 22 and 23) where the MOKE mapping in areas of different thicknesses within one flake was flipped simultaneously (Supplementary Fig. 22e-g). Furthermore, based on the RMCD signals in two FPSe/FGT heterostructures, the non-local exchange bias effect across the whole sample was homogeneous, independent of thickness (Supplementary Fig. S22 and S23). A detailed explanation was also added to the revised manuscript with blue highlights.

Reference

- 1 Fei, Z. et al. Two-dimensional itinerant ferromagnetism in atomically thin Fe₃GeTe₂. *Nat. Mater.* **17**, 778-782 (2018).
- 2 Niu, W. et al. Antisymmetric magnetoresistance in Fe₃GeTe₂ nanodevices of inhomogeneous thickness. *Phys. Rev. B* **104**, 125429 (2021).
- 3 Xia, M. et al. Magnetic Circular Dichroism Study of Electronic Transition in Metal Fe₃GeTe₂. *J. Phys. Chem. C* **126**, 8152-8157 (2022).

The revised contents:

(1) Pages 14-15, lines 306-315:

The single-domain state of FGT was proved in new FPSe/FGT heterostructures (Supplementary Fig. 22a), with the MOKE mappings of the FGT and heterostructure measured at 5 K (Supplementary Fig. 22e-g). The magnetism in areas of different thicknesses within one flake was flipped simultaneously, clearly indicating the single-domain nature of FGT flakes and confirming the sample homogeneity. Furthermore, we measured the non-local exchange bias field in two heterostructures with various thicknesses (sample D and sample E) (Supplementary Fig. 22 and 23). By comparing the RMCD signals of the heterostructure regions with those of the adjacent connected bare FGT region, the non-local exchange bias field across the same sample was

homogeneous.

Supplementary Fig. 22 | RMCD signal of FPSe/FGT heterostructure (sample D). **a** The optical image of FPSe/FGT heterostructure. Positions 1,3, and 5 represent the heterostructure regions, and positions 2,4, and 6 represent the adjacent connected bare FGT regions. **b-d** the RMCD signals of positions 1 and 2 (**b**), 3 and 4 (**c**), and 5 and 6 (**d**), respectively. **e-g** MOKE mappings under the different magnetic fields for the FPSe/FGT heterostructure. All the data was measured at 5 K.

Supplementary Fig. 23 | RMCD signal of FPSe/FGT heterostructure (sample E). **a** The optical image of the FPSe/FGT heterostructure. Positions 1 and 3 represent heterostructure regions, and positions 2 and 4 represent the adjacent connected bare FGT regions. **b, c** the RMCD signals of positions 1 and 2 (**b**), 3 and 4 (**c**), respectively.

4. Since the exchange bias should be an interfacial effect, it is not clear why the FePSe3 thickness should matter as long as it remains in the thin bulk limit. This seems to be confirmed by Supplementary Figure 12, and the lack of any t_{AFM} term in the Mauri model. Yet, the exchange bias in Fig. 3b more than doubles with a change of only 4 nm (from 17.2 nm to 21.6 nm). Can the authors explain this discrepancy?

Response: Thanks for the reviewer's valuable comments. The thickness of the antiferromagnet can also affect the exchange bias effect, as shown in Supplementary Fig. 12, because the device should follow the generalized Meiklejohn-Bean model¹⁻³.

$$H_{EB} = -\frac{A_{12}/\xi}{\mu_0 M_{FM} t_{FM}} + \frac{A_{12}^3}{8K_{AFM}^2 \mu_0 M_{FM} t_{FM} t_{AFM}^2}$$

Where A_{12} is the interfacial exchange stiffness, ξ is the distance of the interlayer, M_{FM} is the saturation magnetization of the ferromagnet, K_{AFM} is the anisotropy constant of the AFM layer, t_{FM} is the FM thickness, and t_{AFM} is the AFM thickness.

In the above equation, the first term represents the exchange bias field with a sufficiently large AFM thickness, which is the Mauri model. The second term varies as $1/t_{AFM}^2$, reveals that the AFM thickness influences the local exchange bias effect, especially for thin AFM thickness conditions where H_{EB} would be more strongly affected by t_{AFM} than t_{FM} . The above model provides the reason that the exchange bias field was enhanced by about 2 times when the thickness of AFM materials changes by 4 nm. Here we have revised our claims more clearly to demonstrate the AFM thickness dependence, as highlighted in blue and shown below.

The revised contents:

(1) Page 8, lines 158-163:

On the other hand, according to the generalized Meiklejohn-Bean model⁴⁸⁻⁵⁰, H_{EB} can be written as:

$$H_{EB} = -\frac{A_{12}/\xi}{\mu_0 M_{FM} t_{FM}} + \frac{A_{12}^3}{8K_{AFM}^2 \mu_0 M_{FM} t_{FM} t_{AFM}^2} \quad (1)$$

Where A_{12} is the interfacial exchange stiffness, ξ is the distance of the interlayer, M_{FM} is the saturation magnetization of the ferromagnet, K_{AFM} is the anisotropy constant of the AFM layer, t_{FM} is the FM thickness, and t_{AFM} is the AFM thickness.

Reference:

- 1 Keller, J. et al. Domain state model for exchange bias. II. Experiments. *Phys. Rev. B* **66**, 014431 (2002).
- 2 Lund, M. S. et al. Effect of anisotropy on the critical antiferromagnet thickness in exchange-biased bilayers. *Phys. Rev. B* **66**, 054422 (2002).
- 3 Malozemoff, A. P. Random-field model of exchange anisotropy at rough ferromagnetic-antiferromagnetic interfaces. *Phys. Rev., B Condens. Matter* **35**, 3679-3682 (1987).

5. *FePSe₃ is a zig-zag AFM. Do the authors expect that there could be a spatially varying spin texture induced in the FGT as a result?*

Response: We thank for the reviewer's valuable comments. Based on the strong interlayer coupling between FGT and FePSe₃ after laser shocking, the FGT in the vdW heterostructure will also exhibit spatially varying spin texture, induced by local zig-zag AFM spin texture in FePSe₃. According to previously reported works¹⁻³, ferromagnetic spin reorientation can be triggered with the help of electron beam induction or local spin direction induction of low-symmetry antiferromagnets, due to the interlayer coupling between antiferromagnets and ferromagnets.

References

- 1 Nolting, F. et al. Direct observation of the alignment of ferromagnetic spins by antiferromagnetic spins. *Nature* **405**, 767-769 (2000).
- 2 Khan, R. A. et al. Magnetic domain texture and the Dzyaloshinskii-Moriya interaction in Pt/Co/IrMn and Pt/Co/FeMn thin films with perpendicular exchange bias. *Phys. Rev. B* **98** 064413 (2018).
- 3 Joly, L. et al. Spin-reorientation in the heterostructure Co/SmFeO(3). *J Phys. Condens. Matter* **21**, 446004 (2009).

Other questions

1. Fig. 1a I and II are never explained or mentioned anywhere in the text. In my opinion, this panel is not very useful to the story and could be removed entirely. Similarly, Fig. 1b I and II are not directly addressed in the text, and Fig. 1b ii gives the impression that the interlayer spacing can be modulated back and forth (modulation and unmodulation). From my understanding, the laser shock pressure is irreversible, therefore this panel is misleading.

Response: We appreciate the reviewer's valuable comments. We have rearranged the figures based on the reviewer's suggestion. Fig. 1a was moved to the revised supplementary information as supplementary Fig. 24 and the detailed explanation for this figure was added to the discussion in the revised manuscript with blue highlighting. This figure shows a prospective diagram of a typical magnetic random-access memory (MRAM). The exchange bias effect is the basis of MRAM, and the related performance improvements can promote MRAM designs with help of the laser shocking strategy.

In addition, we have included more clarifications for Fig. 1b (Fig. 1a after revision) in the manuscript because Fig. 1b is related to the mechanism and our methodology. Fig. 1b I stressed that H_{EB} is inversely proportional to the interlayer spacing in theory. Fig. 1b II stressed that reducing the interfacial spacing of the AFM/FM can induce a stronger exchange bias field and saturation magnetization.

The laser shock pressure is irreversible and the layer spacing is permanently reduced to enhance the interlayer coupling. We have redrawn Fig. 1b II to avoid possible misunderstanding.

The revised contents highlighted in blue:

(1) Page 16, lines 340-345:

Our work has demonstrated the existence of EB effects in vdW magnetic heterostructures and achieved great enhancement of H_{EB} , T_b , and H_c through vdW spacing tuning. It is worthwhile to emphasize that the interlayer coupling of vdW heterostructures can be manipulated promptly over a large area via LS. This provides the potential to promote MRAM with more stable performance and higher storage density (Supplementary Fig. 24).

Magnetic Random Access Memory (MRAM)

spin valve cell based on EB effect

FPS_e/FGT/h-BN/FGT

Smaller interlayer spacing
Higher performance

Supplementary Fig. 24 | Schematic diagram of Magnetic Random Access Memory (MRAM). Unit cell of MRAM is a spin valve device based on EB effect, whose performance can be improved by reducing the layer spacing.

(2) Page 6, lines 95-99:

As shown in Fig. 1a, according to the generalized Meiklejohn-Bean model³⁶, H_{EB} is inversely proportional to the interfacial spacing (Fig. 1a I). Therefore, reducing the interfacial spacing of the AFM/FM heterostructure can induce a stronger EB field and saturation magnetization (Fig. 1a II).

Fig. 1 a | Mechanism and schematic diagram of EB effect enhanced by the reduction of AFM/FM interlayer spacing.

2. I found that several figure panels were repetitive, redundant, or unnecessary. For instance, Fig. 1c, 2a, and 3a are very similar; and Supplementary figure 7 seems unnecessary.

Response: Thanks for the reviewer's valuable comments. We have rearranged the

figures. We have revised Fig. 1c in the revised manuscript as shown below, and removed Fig. 2a, Fig. 3a in the revised manuscript and Supplementary Fig. 7 the supporting information.

The revised parts in blue highlighting in the revised manuscript are listed below.

(1) Page 6-7, lines 92–133:

A typical MRAM based on the EB effect is composed of a matrix of spin valves of vdW heterostructures. To obtain higher performance requirements such as high tunneling magnetoresistance and high read fault tolerance^{5, 7}, the modulation of the EB effect can enhance the interlayer coupling to optimize performance. As shown in Fig. 1a, according to the generalized Meiklejohn-Bean model³⁶, H_{EB} is inversely proportional to the interfacial spacing (Fig. 1a I). Therefore, reducing the interfacial spacing of the AFM/FM heterostructure can induce a stronger EB field and saturation magnetization (Fig. 1a II). Here, FGT and FPSe single crystals were confirmed by scanning electron microscope and energy dispersive spectrometer (Supplementary Fig. 1 and 2), and their bulk magnetizing characteristics are shown in Supplementary Fig. 3, indicating a Néel temperature T_N of 113 K for FPSe and a Curie temperature T_C of 230 K for FGT, consistent with those reported in the literature³⁷⁻⁴⁰. To study the interlayer coupling tuned by vdW spacing engineering, we prepared heterostructures composed of FGT and FPSe flakes. Raman spectrum characterized the FPSe, FGT, and FPSe/FGT heterostructure, where the peaks could well match those of FPSe and FGT, indicating the excellent quality of the FPSe/FGT heterostructure (Supplementary Fig. 4). The LS technology enables the modulation of the interlayer spacing of the vdW heterostructures, which can provide high pressure up to ~ 20 GPa peak level in an ultrashort time scale (tens of picosecond), like a “hammering” operation. Fig. 1b II shows a schematic diagram of the LS process (Method for details). As shown in Fig. 1b, the FPSe/FGT vdW heterostructure with a natural vdW interface distance of d_1 exhibits a weak interlayer coupling. After LS^{41, 42}, its vdW interface distance was reduced to d_2 , promising a stronger interlayer coupling. The evolution of interface spacing under LS was studied by molecular dynamics (MD) simulations (Fig. 1c). The equilibrium states for both the unstrained configuration and strained configuration were simulated for the FPSe/FGT system, showing an interface spacing (distance between flake edge atoms) of 6.6 Å and 4.2 Å before and after LS respectively. (Supplementary Information

Section 3 and Section 4). Supplementary Fig. 5-7 and Table 1-3 show the calculation details and strain evolution of the LS MD simulation. More importantly, LS is effective to flatten wrinkles and voids without any damage to the heterostructures^{43, 44}. The Raman spectrum and mapping before and after LS are shown in Supplementary Fig. 8 and 9, the E_{2g}^2 peak of FGT, and the E_{2g}^1 peak of FGT and A_{1g} peak of FPSe in the FPSe/FGT heterostructure showed negligible change, indicating that FGT and FPSe/FGT heterostructure sustained their good quality after LS. Further, we used cross-sectional (scanning transmission electron microscopy) STEM images to measure the interface gap width of the FPSe/FGT heterostructure. The STEM images of the samples before and after LS were acquired along different orientations because the laser shocking process outside the chamber made it hard to conduct in-situ characterizations. After LS, vdW interlayer distances of FPSe and FGT were fixed at their intrinsic values of ~ 2.5 Å and ~ 2.1 Å respectively, consistent with previous reports^{40, 45-47}, and there was no damage to the lattice structure, but the interface gap width (initial ~ 7.4 Å) was dramatically reduced to ~ 4.6 Å (Fig. 1d, e).

(2) Pages 7-8, lines 136-140:

According to the above mechanism, reducing the interface spacing in AFM/FM heterostructures is expected to significantly enhance H_{EB} and T_b . To confirm the enhancement of H_{EB} and T_b in AFM/FM heterostructures via LS technology, magneto-optical Kerr effect (MOKE) techniques were used to investigate the magnetic properties of a FPSe/FGT heterostructure, denoted as sample A (Supplementary Fig. 10a).

(3) Pages 9-10, lines 185-195:

According to the generalized Meiklejohn-Bean model, H_{EB} can be modulated by both the interlayer spacing of the AFM/FM heterostructure, and the thickness of AFM or FM material. Thickness modulation represents the traditional device performance modulation method by changing the material thickness. Since the thickness cannot be modulated in situ, so new devices need to be prepared. LS modulation represents a novel approach to device performance modulation by changing the interface spacing. LS modulation can be modulated in situ directly on the fabricated devices, showing a more convenient application potential. Equation (1) shows that H_{EB} increases to saturation with the increase of t_{AFM} and decreases with the increase of t_{FM} , therefore, the effect of FM or AFM thickness on H_{EB} for the FPSe/FGT heterostructure was

investigated.

(4) Page 25, Fig. 1:

Fig. 1 | Characterization of vdW spacing of FPSe/FGT heterostructures. **a** Mechanism and schematic diagram of EB effect enhanced by reducing the AFM/FM interlayer spacing. **b** The schematic of the transfer process of FPSe/FGT heterostructure (I) and LS process (II) on a SiO₂/Si substrate ($d_1 > d_2$). Panel II shows a schematic diagram of the laser shocking process. **c** Equilibrium position for a FPSe/FGT system at the unstrained condition (top), strained condition (bottom) in molecular dynamics simulations. **d**, **e** The cross-sectional HAADF-STEM image of FPSe/FGT heterostructure before (**d**) and after (**e**) LS. Inset: intensity profile (right panel) along white dashed line in the cross-sectional HAADF-STEM image (middle panel).

(5) Page 26, Fig. 2:

Fig. 2 | Enhanced EB effect through LS. **a, b** The temperature-dependent Kerr loops of FPSe/FGT heterostructures before and after LS. **c** Kerr loops versus the magnetic field for FPSe/FGT heterostructure through the magneto-optical Kerr test before and after LS at 5 K. **d** A detailed comparative investigation for H_{EB} and T_b versus the temperature before and after LS at 5 K. **e** Summary of H_{EB} and T_b enhancements before and after LS. **f, g** A detailed comparative investigation for H_{C-L} (**f**) and H_{C-R} (**g**) versus the temperature before and after LS at 5 K. **h** R versus the temperature for the FPSe/FGT heterostructure at -180 mT.

Fig. 3 | Comparison of interlayer spacing-modulation and thickness-modulation. **a** Kerr loops for individual FGT (26.9 nm) and FPSe (17.2 nm, 21.6 nm)/FGT (26.9 nm) heterostructures, respectively, measured at 5 K. **b** Kerr loops for individual FPSe (27.0 nm)/FGT (18.0 nm, 25.5 nm, and 22.4 nm) heterostructures, respectively, measured at 5 K. **c** The temperature-dependent Kerr loops of FPSe/FGT heterostructures are measured under 0 Gpa, ~8 Gpa, ~11 Gpa, and ~13 Gpa at 5 K, respectively. **d** The evolution of H_{EB} on t_{FGT} and t_{FPSe} . **e** The evolution of H_{EB} on

temperature and pressure before and after LS. **f, g** The evolution of H_{EB} and T_b on pressure and t_{AFM}/t_{FM} . **h** Schematic diagram of spin valve device. **i** Tunneling resistance of the pristine device (I), thickness modulation device (II), and LS modulation device (III) with B applied parallel to the FGT c -axis. Device structure diagrams are shown in the upper part. Thickness modulation is achieved by constructing a new heterojunction. pristine device and thickness modulation device have different top FGT and FPSe thicknesses. LS modulation reduces the interlayer spacing of pristine device by applying pressure in situ.

3. The authors notation of States 1-3 for the spin valve device was confusing and the bottom panel in Fig. 3i is not useful. It would be clearer to just use “Pristine”, “Thickness Modulated”, and “Laser Shock Modulated”. In fact, it is not clear what the thickness modulation refers to since the thickness cannot be modulated in situ. If it refers to separate devices, the details should be clearly presented. The same is true for comparison between all the different devices. The authors claim significant enhancements, but the paper needs a clearer comparison of the effect the various FGT and FePSe3 thickness can have on the individual device performance.

Response: We are grateful for the reviewer's valuable comments. The schematic was changed to "pristine", "thickness modulated" and "laser shock modulated" as suggested by the reviewer. The revised Fig. 3i is shown in reply to **the above Comment 2**.

The thickness modulation refers to different devices with different thicknesses in our manuscript, which agreed with the reviewer's opinion. Because the thickness cannot be modulated in situ, thickness modulation requires the preparation of different devices. Therefore, we revised our claims to clarify the discussions of comparing different devices more clearly, as highlighted in blue and shown below. We have also revised the discussions to explain the comparison of different devices.

The emphasis on significant enhancement of performance is by comparing the results before and after laser shocking for the same device, which means that regardless of the thicknesses of FPSe or FGT materials, the exchange bias field and coercivity field of the AFM/FM heterojunction are significantly enhanced after laser shocking, this has been proved in spin valves in Fig. 2a, 2b, Fig. 3i (I), (III), and Supplementary Fig. 19. However, there is no quantitative relationship between the coercive field and

the material thicknesses without using laser shocking strategy according to the MOKE results in the revised Fig. 3a and 3b, in accordance with the claims of previous works^{1, 2}. In addition, new spin valves with different material thicknesses (Fig. R1) have been prepared and compared with the performance of the spin valve devices in Fig. 3i(I) and (II) in the manuscript. The results show that there is no significant relationship between the field window of the spin valves and the thicknesses of the magnetic material. Therefore, we can only conclude that laser shocking can improve spin valve performances regardless of the material thickness.

References

- 1 Simon, E. et al. Magnetism and exchange-bias effect at the MnN/Fe interface. *Phys. Rev. B* 98, 094415 (2018).
- 2 Kim, J. et al. Layer thickness dependence of the current-induced effective field vector in Ta|CoFeB|MgO. *Nat. Mater.* 12, 240-245 (2013).

The revised contents:

(1) Pages 11-13, lines 232-266:

Figure 3h shows a schematic diagram of the spin valve device. The pristine spin valve device and the spin valve device modulated by two modulation modes are shown in the upper schematic diagram of Fig. 3i. To demonstrate spin valve behavior, the resistance of the MTJ as a function of a perpendicular magnetic field was measured. Figure 3i shows the results measured at 5 K. For the pristine device (the optical image in Supplementary Fig. 16), as B was swept from negative to positive values, the resistance suddenly increased from approximately ~ 20 to ~ 49 k Ω at ~ 60 mT, followed by a sudden decrease back to ~ 20 k Ω at ~ 80 mT. As the magnetic field was swept back, an analogous abrupt increase and equally abrupt decrease in tunneling magnetoresistance were observed at ~ -80 mT and ~ -120 mT. This was precisely the behavior expected for a tunneling spin-valve due to the hysteretic magnetization switching of two ferromagnetic electrodes. For thickness modulation, a new device (thickness modulation device, the optical image in Supplementary Fig. 17) was constructed to investigate thickness modulation since thickness cannot be modulated in situ, whose thicknesses of top FPSe and FGT were different from that of the pristine device. For LS modulation, LS was processed directly on the pristine device to modulate the

performance. Spin valve behavior was observed after both modulations (Fig. 3i). Supplementary Fig. 18 shows the resistance of the MTJ as a function of magnetic field (B) from 5 K to 120 K. The magnitude of the TMR was defined as $(R_{AP} - R_P)/R_P$, which characterizes the transmission efficiency, where R_{AP} and R_P are the resistance obtained for parallel and antiparallel alignments of the magnetization. The TMR values were 141%, 130%, and 154% at 5 K for the pristine device Fig. 3i (I), thickness modulation device (Fig. 3i (II)), and LS modulation device (Fig. 3i (III)), respectively. The maximum field window of the thickness modulation device and LS modulation device is 3.5 and 7.5 times larger than that of the pristine device, respectively. Compared to thickness modulation, interlayer spacing modulation is more stable and efficient with a reduced cost. To further verify the significant performance enhancement after LS modulation, the tunnel resistance of thickness modulation device was measured after LS, showing a three-fold enhancement (Supplementary Fig. 19). In addition, the TMR and the field window of the FGT/h-BN/FGT spin valve before LS was 119% and 20 mT, respectively, as shown in Supplementary Fig. 20. Compared to FGT/h-BN/FGT spin valve in our work, the field window under LS modulation was around 15 times larger than that of the vertical FGT/h-BN/FGT spin valves at 5 K. The experimental results confirmed the great advantage of LS modulation in realizing high-performance spin valve devices.

Fig. R1 | TMR measurement of different spin valve devices. The tunnel resistance of two FPSe/FGT/hBN/FGT spin valve devices with different thicknesses at 5 K with a magnetic field applied perpendicular to the Fe₃GeTe₂ plane. The bias current is fixed at 10 μ A. Combined with the spin valve devices in the manuscript, the correlation between devices performance and thickness is hard to obtain.

4. Since laser shockwave experiments are relatively unexplored so far in the field of 2D materials, further details and depictions of the experimental setup would be greatly appreciated and would make the paper much more understandable for a general audience. The methods section should therefore be greatly expanded.

Response: We thanks for the reviewer's valuable comments. We have revised the methods section to describe the details of laser shock wave experiment and experimental setup more clearly, as highlighted in blue and shown below.

The revised content:

(1) Pages 17-18, lines 386-412:

Methods

Details of Laser Shock Processing. The FPSe/FGT vdW heterostructures placed on SiO₂ (300 nm) /Si substrate were nano-strained by employing laser shocking pressure. Fig. 1b II shows a schematic of the laser shocking process. First, A SiO₂ (300 nm) /Si substrate with FPSe/FGT heterostructure was placed on a slide and covered with a clean metal film. Here, the metal film was a 10 μ m aluminum (Al) foil, which functioned as a buffer layer and a contaminant barrier due to its excellent ductility and low cost. Further, nano-graphene powder (Hangdan Optoelectronics Technology, China) was uniformly spun onto Al foil as an ablative coating with a thickness of approximately 1.0 mm. Note that the range of the spin-coated graphene powder needs to completely cover the SiO₂ (300 nm) /Si substrate area. A cover glass was then placed over the graphene powder to serve as a shock confinement layer. In this way, the FPSe/FGT heterostructure was successfully sandwiched between two glass sheets. Finally, multiple laser pulses were implemented by placing the prepared sample on an X–Y–Z computer-controlled motorized stage. The laser position was fixed by adjusting the laser focus. And the step size and path of the X-Y-Z electric platform were set to realize the

pressure application in the customizable area.

Next, a pulsed (10 ps) Q-switch Nd:YAG laser (Continuum Surelite III, Wavelength: 1064 nm, Power: 0.3 ~ 1.4 GW/cm²) was used as an energy source for ablation. The laser beam diameter of 4 mm was attained by a focusing lens, which was calibrated by a photosensitive paper (Kodak Linagraph, type: 1895). The laser pulse irradiated the ablative coating (1.0 mm graphene powder). Shock waves were generated by sublimation of the ablative layer using laser pulses. Due to the constraints of the confinement layer, the shock wave could propagate through the Al foil and apply pressure at the nanoscale resolution on the SiO₂ (300 nm) /Si substrate with FPSe/FGT heterostructure. The pressure depends on the intensity of the laser and the duration of the pulse during the laser shocking process.

Reviewers' Comments:

Reviewer #2:

Remarks to the Author:

The authors have entirely addressed my concerns with new data and clearer text and explanations. I applaud their efforts and strongly support publication of this highly interesting work in Nature Communications.

Response Letter

We sincerely thank you and the reviewers for the great efforts providing valuable comments concerning our manuscript entitled “Manipulating Exchange Bias in 2D Magnetic Heterojunction for High-performance Robust Memory Applications” (ID: NCOMMS-22-35123A). The comment is helpful for revising and improving our paper. We have fully addressed the comment carefully. The following is the point-to-point response to the comment.

Reviewer #2

The authors have entirely addressed my concerns with new data and clearer text and explanations. I applaud their efforts and strongly support publication of this highly interesting work in Nature Communications.

Response: We are grateful for the reviewer’s comments and appreciation for our work.